# Fast and Accurate Sperm Detection Algorithm for Micro-TESE in NOA Patients

**DOI:** 10.3390/bioengineering12060601

**Published:** 2025-05-31

**Authors:** Mahmoud Mohamed, Konosuke Kachi, Kohei Motoya, Masashi Ikeuchi

**Affiliations:** 1Laboratory for Biomaterials and Bioengineering, Institute of Science Tokyo, Tokyo 101-0062, Japan; mahmoud.mohamed.abda@tmd.ac.jp; 2Graduate School of Information Science and Technology, University of Tokyo, Tokyo 113-0033, Japan; konosuke_kachi@ipc.i.u-tokyo.ac.jp; 3Sanamedi, Inc., #601 Nihonbashi Life Science Building, 2-3-11 Nihonbashi-Honcho, Chuo-ku, Tokyo 103-0023, Japan; motoya@sanamedi.jp

**Keywords:** sperm detection, Micro-TESE, non-obstructive azoospermia, DIC microscopy, classical image processing, biomedical simulation

## Abstract

Purpose: Non-obstructive azoospermia (NOA) presents major challenges in assisted reproductive technology (ART) due to the extremely low number of viable sperm within testicular tissue. In Micro-TESE procedures, embryologists manually search for sperm under DIC microscopy—a slow, labor-intensive process. We aim to streamline this process with an efficient computational detection tool. Methods: We present SD-CLIP (Sperm Detection using Classical Image Processing), a lightweight, real-time algorithm that simulates sperm structure detection from unstained DIC images. The model first identifies convex sperm head candidates based on shape and width using edge gradients, then confirms the presence of a tail via principal component analysis (PCA) of pixel clusters. Results: Compared to the MB-LBP + AKAZE method, SD-CLIP improved processing speed by 4× and achieved a 3.8× higher posterior probability ratio, making detected sperm candidates significantly more reliable. Evaluation was performed on both human Micro-TESE and mouse testis images, demonstrating robustness in low-sperm environments. Conclusions: SD-CLIP simulates a domain-specific image interpretation model that identifies sperm morphology with high specificity. It requires minimal computational resources, supports real-time integration, and could be extended to automated sperm extraction systems. This tool has clinical value for accelerating Micro-TESE and increasing success rates in ART for NOA patients.

## 1. Introduction

Infertility affects about one in six individuals globally, with an estimated lifetime prevalence of 17.5% (95% CI: 15.0–20.3), indicating a significant public health concern [1]. Infertility can be broadly classified into two types: female-derived infertility and male-derived infertility, with male-related causes accounting for 40–50% of infertility cases [2]. Among the male-factor causes, non-obstructive azoospermia (NOA) stands out due to its severely impaired or absent sperm production, making assisted reproductive treatment particularly challenging [3]. In such cases, techniques like testicular sperm extraction (TESE), and especially microdissection TESE (Micro-TESE), are employed to locate and retrieve sperm directly from the seminiferous tubules [4]. While Micro-TESE can improve sperm retrieval rates and minimize tissue damage, embryologists often spend considerable time manually searching under a microscope—an effort that can be stressful for both the patient and the clinical team. The average time for successful Micro-TESE is 1.8 h, the average time for unsuccessful Micro-TESE is 2.7 h, and the maximum time is reported to be 7.5 h [5]. Because sperm in NOA tend to be extremely sparse or immotile, they are difficult to distinguish from other cells (e.g., Sertoli cells and spermatogonia) found in the seminiferous tubules [6]. The long operation time is a heavy burden on the embryologist and patient, and the possibility of missing good sperms due to the lack of concentration cannot be overlooked. In other words, the efficiency of sperm retrieval is an important issue in Micro-TESE, and an efficient sperm retrieval system would contribute to faster Micro-TESE and higher success rates.

To address this detection problem, computational methods leveraging artificial intelligence (AI) and advanced image-processing techniques have been developed [7,8]. These approaches can streamline the search process, but they often rely on large training datasets or can be computationally intensive. Additionally, other methods such as microfluidic sorting or free-flow acoustophoresis target motile sperm based on physical properties, which may not be suitable for immotile sperm frequently encountered in NOA [9,10].

In contrast, differential interference contrast (DIC) microscopy allows clear, high-contrast imaging of unstained living cells [11]. This capability is especially helpful for identifying subtle morphological features of sperm. Building on classical image processing—which can be faster and more resource-efficient than typical deep-learning solutions [7]—we propose Sperm Detection using Classical Image Processing (SD-CLIP), a two-step algorithm tailored for Micro-TESE. First, candidate sperm heads are identified using a filter specialized for the convex shape and specific width of sperm heads; next, the presence of a tail is confirmed via principal component analysis (PCA).

In this paper, we detail the design and implementation of SD-CLIP and compare it against an established MB-LBP-based method [8]. We apply our algorithm to mouse seminiferous tubule samples as a proxy for human tissue and conduct performance evaluations in terms of detection speed, false positive rates, and overall accuracy. Our results indicate that SD-CLIP markedly accelerates sperm detection—reducing computational time approximately fourfold—and substantially lowers the false positive rate, making it especially useful for low-sperm environments characteristic of NOA. The algorithm’s reliance on fundamental image-processing operations also facilitates potential real-time application and integration with automated pipetting systems for more efficient and less operator-dependent Micro-TESE procedures.

## 2. Materials and Methods

### 2.1. Differential Interference Microscopy

To observe cells in their living, unstained state, it is more suitable to use microscopy techniques that utilize the phenomenon of light interference, such as DIC microscopy, rather than conventional bright-field microscopy.

In Micro-TESE, it is necessary to detect sperm without staining for use in fertilization. Therefore, sperm detection is conducted using images observed with a differential interference microscope, which is suitable for unstained observation. In this study, differential interference observation was carried out using the inverted microscope IX70-DIC (Olympus, Tokyo, Japan). Details of the optical system of the microscope IX70-DIC used are as follows:Objective lens: UPlanFL10x NA0.30 ∞/− (10× magnification, semi-apochromat lens);Eyepiece lens: WH10x/22, 2 pieces (10× magnification, field number 22, one for each eye);Filter: Color temperature conversion filter + ND filter + Frost filter;Illumination: 100 W halogen transmission lighting.

For the sperm detection software to be used in MD-TESE clinically, it is required to balance accuracy and speed at a clinically practical level. Therefore, this study employed an algorithm that utilizes the features of images obtained with a differential interference microscope while keeping the computational load low.

### 2.2. Detection Using MB-LBP

Sasaki et al. [8] proposed a method of using facial recognition algorithms for sperm detection. In addition to the methods proposed for facial recognition, they have reduced computational costs by narrowing down sperm candidates through feature point extraction using AKAZE features [12]. This method employs the Multi-block Local Binary Pattern (MB-LBP) to encode the surrounding information of a focused pixel into a binary pattern.

However, since AKAZE is a general feature extraction algorithm, it is not specialized for sperm candidate detection. Therefore, replacing sperm candidate detection using AKAZE with a method specialized for this purpose could potentially speed up detection and reduce the sperm detection time by decreasing the number of candidates. Furthermore, introducing a decision algorithm specialized in sperm determination could also reduce computational requirements.

To address these issues, we propose a new sperm detection algorithm. This new algorithm, similar to the detection method using MB-LBP, first detects sperm candidates from the entire image and then performs sperm detection on each candidate to achieve faster processing. The candidate detection algorithm employs convolution with a unique filter, specialized for sperm candidate detection, aiming to reduce both computation time and the number of candidates. A new algorithm specialized in sperm determination is developed for application to each sperm candidate.

### 2.3. Proposed SD-CLIP Algorithm

#### 2.3.1. Sperm Head Shape-Based Candidate Detection

To reduce computational effort, the initial step involves detecting candidates for the sperm head across the entire image, followed by a final determination of whether these candidates are indeed sperm based on the presence or absence of a tail. Human sperm size and shape are relatively consistent across individuals. Utilizing this, spherical structures of a certain size present in the image are considered candidates for the sperm head.

In Micro-TESE, a section of the seminiferous tubule is collected, containing not only sperm but also other cells like spermatocytes, which often have a spherical shape. Therefore, most parts inside these cells are convex, with concavities primarily at the cell boundaries (Figure 1).

DIC microscopy can be considered a device that converts differential information into brightness for visualization. Here, the brightness value *I*(x, y) in the image is thought to be proportional to the energy of the interference light *Jδθ*(*x*, *y*). By setting the amount of bias retardation between the two polarized lights separated by the Wollaston prism to −90 degrees, a rough proportional relationship exists between the height *h*(*x*, *y*) from a reference point, and the brightness value *I*(*x*, *y*), using a constant C, assuming the x-axis is oriented in the direction in which light is split inside the differential interference microscope:(1)$$I\sim -\frac{\partial h}{\partial x}+C$$

By adjusting variables and constants to make the proportionality coefficient equal to 1, this equation can be rewritten as follows:(2)$$I=-\frac{\partial z}{\partial x}+C^{\prime }$$
where *z*(*x*, *y*) is a variable in the height direction, and *C*′ is a constant. Assuming the average gradient in the *x* direction, *∂z/∂x*, across the entire microscopic image equals zero, $C^{\prime }=\bar{I}$ holds true, leading to the following:(3)$$-\frac{\partial z}{\partial x}=I-\bar{I}$$

Differentiating both sides of the equation, with respect to *x*, yields the equation for determining the curvature in the *x* direction, $\partial^{2}z/\partial x^{2}$:(4)$$-\frac{\partial^{2}z}{\partial x^{2}}=\frac{\partial I}{\partial x}$$

From these equations, it is clear that the gradient in the x-direction can be derived by subtracting the average brightness value from the brightness values, and the curvature in the x-direction can be obtained from the gradient of the brightness values in the x-direction.

The gradient of brightness values in the x-direction can be approximately determined using the Sobel filter. Furthermore, as mentioned earlier, since many parts of the observed cells’ interiors are convex and the concavities are mainly at the cell boundaries, pixels that become negative through the Sobel filter [13], indicating areas where the curvature in the x-direction ($\partial^{2}z/\partial x^{2}$) is negative, can be considered candidates for cell edges. Moreover, by utilizing the fact that edges with a positive gradient in the x-direction (∂z/∂x) are on the x-negative side of the cell (left side in Figure 2), and those with a negative gradient are on the x-positive side (right side in Figure 2), it is understood that edges with brightness values below the average brightness are on the x-negative side of the cell, and edges with brightness values above the average are on the x-positive side.

Regarding image processing steps, initially, color images captured by the camera mounted on the differential interference microscope are converted into grayscale images (where each pixel value represents brightness). Subsequently, a Gaussian filter is convolved with this image to obtain a smoothed image, denoted as $I_{b}$. Convolution with a Gaussian filter smooths the image by removing noise that causes abrupt changes in brightness. Then, applying an x-axis directional Sobel filter to the smoothed image $I_{b}$ produces a two-dimensional array denoted as $G_{x}$. The comparison and bitwise operations between two-dimensional arrays are performed element-wise. Expressing the edge detection,(5)$$edge=\left(G_{x}<\tau \right)$$(6)$$edge\text{\_}right=edge\cap \;\left(\hat{I}>\bar{I}\right)$$(7)$$edge\text{\_}left=edge\cap \;\left(\hat{I}\leq \bar{I}\right)$$
where τ is an appropriate value serving as the threshold for edge detection. In terms of x-axis processing, the implementation is as follows:(8)$$\tau =-sn\times SD\left(\frac{\partial^{2}z}{\partial x^{2}}\right)$$

Here, SD represents the function calculating the standard deviation, and sn is a modifiable scale factor within the range [0.1, 2.0].

From this edge information, to detect cells with a certain width w as sperm head candidates, one should take the bitwise AND of two arrays: one shifted w units in the negative direction from the positive edge, and the other shifted w units in the positive direction from the negative edge (Figure 3A–C). Using NumPy notation, this can be expressed as follows:(9)$$CANDIDATE=edge\text{\_}left\left[:-w\right]\cap edge\text{\_}right\left[w:\right]$$

CANDIDATE forms several clusters. By applying the Block-Based Decision Table (BBDT) algorithm [14] and the Scan Array Union Find (SAUF) algorithm [15] to these clusters, it is possible to label connected components and calculate the area of each cluster. Currently, clustering is performed using the connectedComponentsWithStats function from OpenCV (which uses algorithms like BBDT and SAUF). Clusters with a small area are likely to be noise, so clusters with an area above a certain threshold are considered as final sperm head candidates, and their centroids are used as the position of the head (Figure 3D).

While the actual sperm head is not spherical but elliptical in shape, using the method described above alone can detect sperm heads oriented vertically in the image but not those oriented horizontally (Figure 4). Therefore, improvements are made to enable the detection of horizontally oriented sperm as well.

The edge detection described earlier was performed using the value obtained by applying the Sobel filter in the x-direction to the brightness values *I_b_* (*x*, *y*) of the image. Consider applying the Sobel filter in the y-direction instead of the x-direction. This is equivalent to using the first derivatives in both x and y directions instead of the second derivative in x:(10)$$\frac{\partial^{2}z}{\partial x^{2}}\Rightarrow \frac{\partial^{2}z}{\partial x\partial y}$$

This operation allows for the detection of diagonal edges, with the direction (left or right diagonal) determined by the sign (Figure 5). By employing this method for detecting diagonal edges, combined with the edge detection in the x-axis direction, it becomes possible to address sperm head orientations at angles of −π/3 ± π/6, 0 ± π/6, and π/3 ± π/6.

#### 2.3.2. Sperm Determination

For each sperm candidate detected, surrounding pixels are cropped from the image for sperm determination, which is performed based on the presence or absence of a tail. This section explains the details of the algorithm used to determine the presence of a tail.

Although this explanation focuses on vertical sperm determination (corresponding to edge detection in the x-axis direction), the algorithm supports sperm tail detection in arbitrary directions by using rotation alignment, as explained below.

Initially, an image region centered around the centroid of the sperm head candidate is automatically cropped. Then, within this cropped image, pixels considered to be inside some structure (rather than at its edge) are set to 1 (True), and all other pixels are set to 0 (False). For example, in the x-axis edge detection, areas where curvature is negative were used as edges, but for vertical sperm determination based on these results, areas with positive curvature are considered as pixels where cells exist. Subsequently, noise is removed by performing an AND operation on horizontally adjacent pixels.

For sperm head candidates extracted based on x-axis edge detection, the tail can exist in any direction around the head. To support consistent tail detection, our implementation uses principal component analysis (PCA) to estimate the dominant tail direction and rotates the cropped region such that the tail aligns vertically. This allows tail detection to be performed uniformly as in the original vertical case. Figure 6 illustrates how PCA is applied to the tail candidate cluster to compute PC1 and PC2 axes. In non-vertical cases, the entire region is rotated using the PC1 angle to normalize the tail orientation.

From the rotated image, only the upper part is cropped (as in the vertical case, corresponding to the magenta rectangle in Figure 6). Then, the BBDT and SAUF algorithms are used to label clusters. If the tail exists in the upward direction after alignment, at least one pixel in the top row of the cropped image must belong to a non-background cluster. If this condition is not met, the candidate is discarded.

PCA is also used to validate whether the cluster is sufficiently “tail-like”. Specifically, the first principal component (PC1) corresponds to the long axis of the tail cluster, while the second principal component (PC2) corresponds to the short axis. A real tail is expected to be elongated along PC1 and narrowly distributed along PC2.

To quantify this, we calculate the variance in the PC2 direction using the corresponding eigenvalue. If this variance is below a certain threshold, the cluster is accepted as a tail. In practice, if the variance along PC2 is below a fixed threshold (6 pixels^2^), the cluster is considered sufficiently elongated and is accepted as a valid sperm tail. This PCA + rotation mechanism was applied throughout the final implementation to support arbitrary tail directions and enable robust detection in real-world DIC images.

### 2.4. Sperm Detection Evaluation Experiment Using Simulated TESE Samples

This study aims to accelerate Micro-TESE for humans. However, images of human seminiferous tubule tissue are difficult to obtain. Therefore, to evaluate the proposed sperm detection method, Micro-TESE is performed on a mouse testis, and the method is applied and evaluated on the obtained seminiferous tubule tissue.

The procedure for Micro-TESE using a mouse testis is described below. First, the testis and surrounding tissues previously extracted from a mouse and stored frozen are thawed and transferred to a Petri dish (Figure 7). To prevent drying, Phosphate Buffered Saline (PBS) is added to the Petri dish containing the tissues.

The Petri dish is placed under a stereo microscope, and the following procedure for collecting testicular tissue is performed. Initially, while holding the testis steady with tweezers to prevent it from rolling, an incision is made with a micro scalpel. Pressing and spreading the incision with tweezers exposes the seminiferous tubules inside to the outside. In patients with NOA, the areas of the seminiferous tubules that may have spermatogenic function are limited. Therefore, under the microscope, seminiferous tubules that are relatively thicker and more turbid are selected for collection. However, in this experiment, mice with normal spermatogenic function are used, meaning many parts of the seminiferous tubules possess spermatogenic function. Thus, an appropriate section of the seminiferous tubule is cut out and collected.

The excised seminiferous tubules are transferred to another Petri dish containing PBS solution to prevent mixing with other tissues. Afterwards, using tweezers and a micro scalpel, the seminiferous tubules are cut into about 10 fragments. The internal tissue from the fragmented seminiferous tubules leaks out into the PBS solution, causing it to become cloudy. Sperm-containing seminiferous tubule tissues are obtained by collecting the clouded areas with a pipette.

A pseudocode-style summary of the SD-CLIP algorithm’s main steps is included in Figure 8.

## 3. Results

### 3.1. Results of Sperm Detection Evaluation Experiment Using Simulated TESE Samples

To evaluate the proposed algorithm, a comparison was made with the method using AKAZE features and MB-LBP (Sasaki et al. [7], hereafter referred to as “Sasaki et al.’s method”), which performs the final determination after detecting sperm candidates, similar to the proposed method. Therefore, the evaluation first focuses on candidate detection in terms of execution time and accuracy. Since the accuracy of sperm detection in the proposed method depends on the shape of the head, candidate detection was evaluated using images taken from human Micro-TESE.

An ASUS Zenbook3 was used for measuring the execution time of candidate detection. The implementation was carried out in Python 3.6.9, using OpenCV 4.1.2 and NumPy 1.17.4 as libraries. Since functions executed internally by OpenCV and NumPy are implemented in compiled languages, they can process data faster compared to functions implemented solely in Python. The calculation of AKAZE features used in Sasaki et al.’s method for sperm candidate detection utilized the detect member function of OpenCV’s AKAZE class. The calculation for sperm candidate detection in the proposed method was implemented with NumPy.

The execution time of the function corresponding to sperm candidate detection was measured by running both Sasaki et al.’s method and the proposed method on 32 microscope images (resolution: 1920 × 1080) taken from human Micro-TESE. The average execution time for each method showed that the proposed method improved the average execution speed by about four times compared to Sasaki et al.’s method. The proposed method took about 70 ms to detect sperm candidates in a 1920 × 1080 microscope image. However, since the algorithm of the proposed method uses only local information in the image for sperm detection, it is easy to parallelize.

Furthermore, the proposed sperm detection algorithm is composed of simple operations such as bitwise and comparison operations for each pixel. Therefore, there is a possibility of further acceleration by utilizing General-Purpose computing on Graphics Processing Units (GPGPU).

Subsequent analysis was conducted on the keypoints detected through sperm candidate detection by each method. Across all 32 microscope images from which keypoints were derived, the proposed method yielded a number of detection candidates that was approximately 1/40th of that by the traditional method (Figure 9).

An example of the detected keypoints is shown in Figure 10. The AKAZE features used in Sasaki et al.’s method are general-purpose features, which is why it can be observed that characteristic points are widely extracted. In contrast, the proposed method specializes in detecting cells of a specific width, resulting in fewer detected candidate points, and it is noticeable that these are generally limited to areas where smaller cells are present.

From the results above, it can be said that the sperm candidate detection of the proposed method is superior to Sasaki et al.’s method in both execution speed and the false positive rate of candidates. Since the sperm determination algorithm that follows sperm candidate detection does not depend on the sperm candidate detection, it is possible to replace the traditional sperm candidate detection algorithm with the proposed method. It is believed that replacing it could result in an improvement in execution speed by one to two orders of magnitude.

Next, the sperm determination algorithm is evaluated using seminiferous tubule tissue obtained from a mouse testis. Mouse sperm differ morphologically from human sperm: they have falciform (hook-shaped) heads and typically longer, more curved tails. In contrast, human sperm tends to have rounded or slightly elliptical heads and straighter tails. These morphological differences may reduce detection accuracy, as the SD-CLIP algorithm was optimized based on human sperm. Therefore, in evaluating the sperm determination algorithm, only the keypoints detected by the proposed method’s sperm candidate detection are treated as the dataset, and accuracy metrics are calculated based on these keypoints alone.

For the detected keypoints, surrounding images of each keypoint are cropped and passed to the sperm determination program (Figure 11). The sperm determination program assesses whether the cell located at the central pixel of each input surrounding image is a sperm and returns the result. To evaluate the sperm determination algorithm, a manual verification of each output was conducted by human observers, and the results were compiled (Table 1).

### 3.2. Performance Comparison with Existing Methods

In clinical scenarios such as Micro-TESE for NOA patients, minimizing false positives is critical, as embryologists manually inspect each detected sperm candidate. A high number of false positives increases workload and may cause viable sperm to be overlooked. Hence, we designed SD-CLIP to prioritize a high true negative rate (TNR), even if it results in a moderate true positive rate (TPR).

In [7], a graph was presented from which we estimated TPR to be 87% and FPR to be 30% for Sasaki et al.’s method (MB-LBP). Using these estimates and the reported data (8671 positive sperm detections and 39,596 negative detections), we created a comparative dataset (see Table 1). Our proposed Sperm Detection using Classical Image Processing (SD-CLIP) achieved a TPR of 58% and a TNR of 95%, whereas Sasaki et al.’s method showed a TPR of 87% and a TNR of 70% (Table 2). Because the TPR and TNR for Sasaki et al. are derived from a published graph rather than raw data, they should be regarded as approximate.

#### 3.2.1. Confusion Matrix and Posterior Probability Ratio (PPR)

To further highlight performance under low-prevalence conditions—typical in NOA—we introduce the Posterior Probability Ratio (PPR), also referred to as a likelihood ratio. Table 3 shows the confusion matrix for the proposed method (SD-CLIP) on a representative test set.

From this matrix, the PPR is calculated as follows:(11)$$PPR=\frac{\frac{TP}{TP+FN}}{\frac{FP}{FP+TN}}$$

Plugging in the numbers:(12)$${PPR}_{SD-CLIP}=\frac{\frac{83}{83+59}}{\frac{27}{27+486}}=11.1$$

Meanwhile, Sasaki et al. (MB-LBP) have an estimated PPR of 2.9, making SD-CLIP approximately 3.8 times more effective in distinguishing true sperm from non-sperm under these highly challenging conditions.

#### 3.2.2. Relevance to NOA: TPR–FPR Trade-Off

Because TPR (sensitivity) and TNR (specificity) typically involve a trade-off, it can be difficult to declare a single “best” method without considering the intended application. In Micro-TESE for non-obstructive azoospermia (NOA), an essential criterion is the probability that a cell deemed “sperm” is indeed sperm—in other words, the positive predictive value (PPV). Let $p_{s}$ represent the proportion of true sperm among the cells being evaluated, $p_{t}$ the TPR, and $p_{f}$ the FPR. Then the PPV p is obtained by the following:(13)$$p=\frac{p_{s}p_{t}}{p_{s}p_{t}+\left(1-p_{s}\right)p_{f}}=\frac{p_{s}p_{t}}{p_{f}+\left(p_{t}-p_{f}\right)p_{s}}$$

Since patients with non-obstructive azoospermia have almost no spermatogenic function, the proportion of sperm in the seminiferous tubule tissue cells is extremely low ($p_{s}\ll 1$). Therefore, unless $p_{f}$ is sufficiently close to 0, it holds that $p_{s}\ll p_{f}$, and the above probability can be approximated as follows:(14)$$p≅\frac{p_{s}p_{t}}{p_{f}}$$

Here, since $p_{s}$ is a constant unrelated to the performance of the sperm determination method itself,(15)$$p\sim \frac{p_{t}}{p_{f}}$$

Since $p_{s}$ (the actual fraction of sperm) is nearly constant but extremely small, the ratio $p_{t}/p_{f}$ strongly determines how often predictions of “sperm” are right. For our proposed method, 0.58/0.053≈10.9, whereas for the traditional MB-LBP, 0.87/0.30≈2.9. Whether we call this ratio “$p_{t}/p_{f}$” or “PPR”, the conclusion remains the same: SD-CLIP outperforms the conventional approach by a factor of about 3.8 in NOA contexts.

Hence, both the confusion matrix-based PPR and the TPR–FPR ratio confirm that our method provides superior accuracy for these low-prevalence, high-stakes scenarios, ensuring that identified sperm are far more likely to be truly viable cells.

### 3.3. Justification of Component Contribution in SD-CLIP

While we were unable to conduct full quantitative ablation experiments due to resource constraints, we provide here a qualitative justification of each major component in the SD-CLIP pipeline to clarify its contribution to detection performance:Edge-Based Candidate Detection:

This initial stage isolates convex structures based on gradient behavior in DIC images. Without it, the algorithm would have no directional filtering mechanism, resulting in many irrelevant regions being marked as candidates.

2.Width-Based Alignment Filtering:

This step ensures the left and right edges correspond to structures matching expected sperm head dimensions. Skipping this would allow arbitrary curved shapes to be included, especially elongated or overlapping cells.

3.Area-Based Noise Removal:

Small or fragmented edge clusters often arise from optical noise or background irregularities. Removing this step would increase false positives by admitting many such spurious components.

4.PCA-Based Tail Verification:

This is the most critical refinement layer. Spherical cells or debris may pass all previous filters, but only real sperms exhibit a distinct narrow tail extension. Omitting PCA-based analysis would result in these non-sperm cells being misclassified, dramatically reducing specificity.

In summary, each module in SD-CLIP is designed to address a specific ambiguity common in Micro-TESE DIC imagery. Removing any one of them would likely reduce overall precision and increase the manual verification burden in clinical use. We include this component-level rationale as an alternative to formal ablation results.

## 4. Conclusions and Future Work

In this study, we presented a new sperm detection algorithm specifically tailored for Micro-TESE in patients with NOA. By integrating classical image processing techniques with DIC microscopy, our method focuses on identifying the sperm head as an initial candidate and then confirming the presence of a sperm tail. Through comprehensive experiments using both human Micro-TESE images and mouse seminiferous tubule samples, the proposed approach demonstrated the following key advantages:Reduced candidate points and faster processing:By leveraging domain-specific information—namely, the approximate size and convex shape of sperm heads—our candidate detection step markedly decreased the number of false positive keypoints compared to conventional methods. Execution time for candidate detection was improved by approximately fourfold, and the algorithm remains amenable to further acceleration via parallelization or general-purpose GPU computing.High specificity in low-sperm environments:Although our approach has a modest TPR relative to a previously proposed method, it achieves a substantially lower FPR. Under the extremely low sperm density conditions characteristic of NOA, the resulting positive predictive value (PPV) of our method is significantly higher. Specifically, the ratio of TPR to FPR—a crucial metric in NOA contexts—showed that our method was roughly 3.8 times more effective than the conventional method.Applicability to Clinical Micro-TESEBy detecting sperm swiftly and minimizing the burden on embryologists, our approach has the potential to shorten operative times and reduce patient discomfort. The algorithm’s reliance on fundamental image-processing operations also facilitates integration into existing clinical workflows with minimal hardware requirements.

Moving forward, future research should focus on refining the trade-off between TPR and FPR while maintaining or enhancing processing speed. This could involve the following:Adaptive Parameter Tuning:Developing automated methods for selecting optimal thresholds in the Sobel filtering and principal component analysis (PCA) steps to further balance sensitivity and specificity.Future versions of SD-CLIP could benefit from using the Scharr operator, which offers improved rotational symmetry and edge accuracy compared to Sobel.The current comparison in Table 2 is based on fixed threshold settings optimized for specificity. A full ROC analysis was not performed due to limited positive cases and data diversity. Future work will include ROC-based evaluation to enable more rigorous comparative benchmarking.Edge misalignment caused by tilted imaging planes or irregular lighting can reduce the consistency of head detection. Although diagonal edge gradients are already included, future implementations could benefit from affine image registration or rotation-normalization preprocessing.Advanced Parallelization:Implementing real-time parallel processing pipelines on GPU or specialized hardware (e.g., FPGA) for speed-critical clinical scenarios.Hybrid Approaches:A potential hybrid system could integrate a lightweight neural network classifier after candidate extraction by SD-CLIP. For instance, once bounding boxes are generated from edge and shape features, a shallow CNN could be used to validate sperm presence based on learned appearance. This modular design would allow robust generalization to diverse imaging conditions while maintaining processing speed.Automation of Sperm Collection:Integrating our detection framework with a robotic arm and pipette system to automatically collect the detected sperm. Such an automated system would further reduce the manual workload on embryologists and could enhance the speed and precision of sperm retrieval in Micro-TESE procedures.

Overall, the proposed algorithm represents a promising step toward faster, more reliable sperm detection in Micro-TESE procedures for NOA patients. By continuing to improve both the detection algorithm and the eventual automation of sperm collection, we aim to increase the likelihood of finding viable sperm and improving outcomes in assisted reproductive technologies.

## Figures and Tables

**Figure 1 bioengineering-12-00601-f001:**
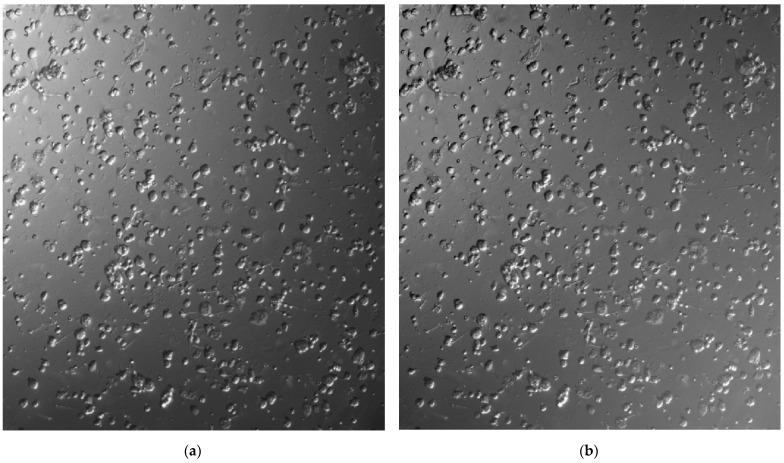
An example of a microscope image obtained by differential interference microscopy (**a**) and the image after brightness correction (**b**). Ideally, the background color of the microscope image should be uniform, but gradients in the background color occur due to external factors such as warping of the container.

**Figure 2 bioengineering-12-00601-f002:**
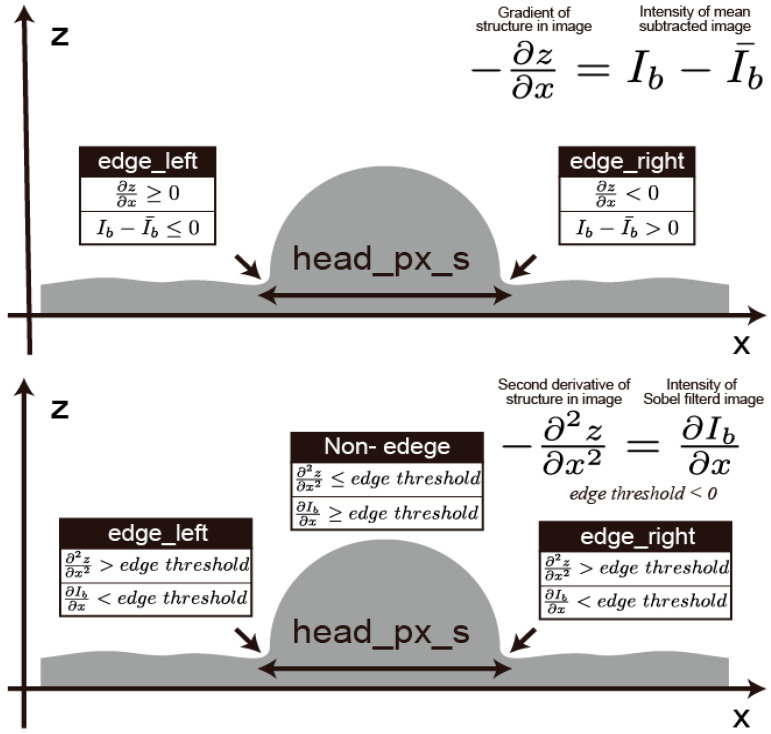
The relationship between the structure of sperm head candidates in the x-axis direction and the brightness values of images taken with differential interference microscopy.

**Figure 3 bioengineering-12-00601-f003:**
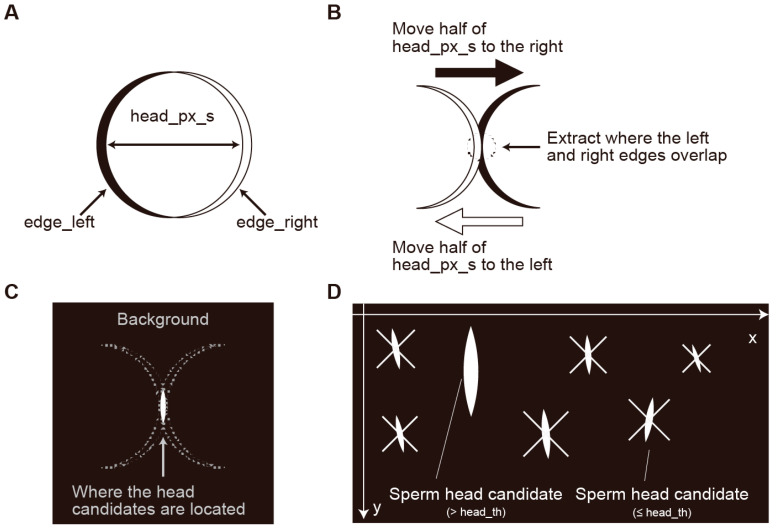
The procedure for determining sperm head candidates based on the left and right edges of structures in an image taken with differential interference microscopy (only for the x-axis direction). (**A**) A schematic diagram showing the left and right edges separated by a predefined width (in the figure, denoted as head_px_s). (**B**) A schematic diagram of the procedure to determine whether there is an overlap when the left and right edges are each moved by half of the predefined width. (**C**) A schematic diagram of the image state after the procedure described in (**B**). The white areas in the figure represent what is referred to as CANDIDATE in the text. In the implementation, only the CANDIDATE areas are 1 (True), while the rest of the areas (Background) are 0 (False). (**D**) A schematic diagram of the procedure to extract areas exceeding a certain threshold (in the figure, denoted as head_th) from within the CANDIDATES. head_th = 20 pixels.

**Figure 4 bioengineering-12-00601-f004:**
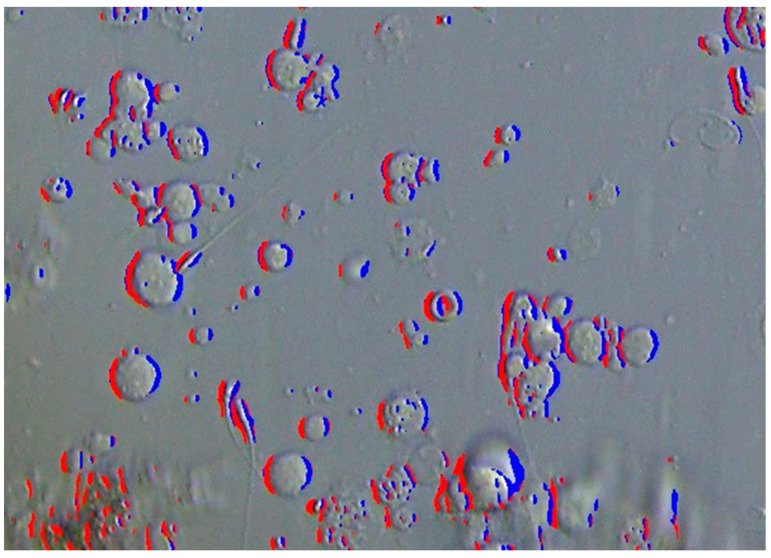
Overlay of areas with negative curvature on the original image. Areas with positive gradients are shown in red, and those with negative gradients are shown in blue, corresponding to the left and right edges, respectively.

**Figure 5 bioengineering-12-00601-f005:**
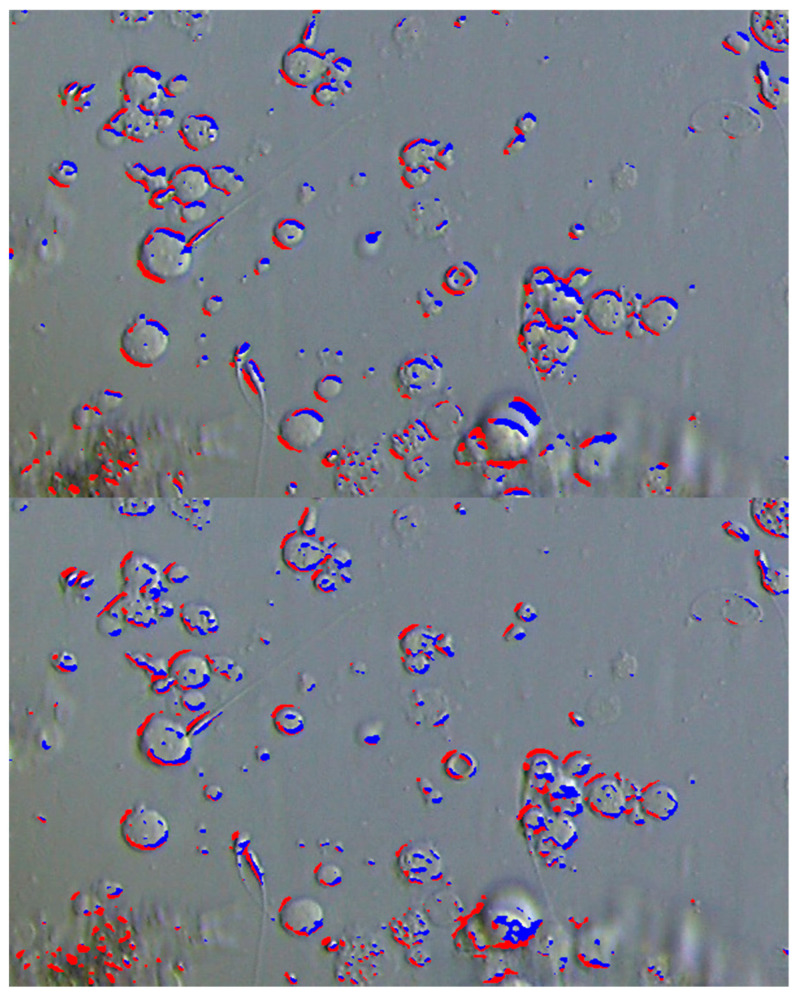
The results when changing the edge detection from ∂z^2^/∂x^2^ to ∂z^2^/∂x∂y (**top**) and to −∂z^2^/∂x∂y (**bottom**). Diagonal edges can be observed. Similar to Figure 4, areas with a positive gradient are shown in red, and those with a negative gradient are shown in blue.

**Figure 6 bioengineering-12-00601-f006:**
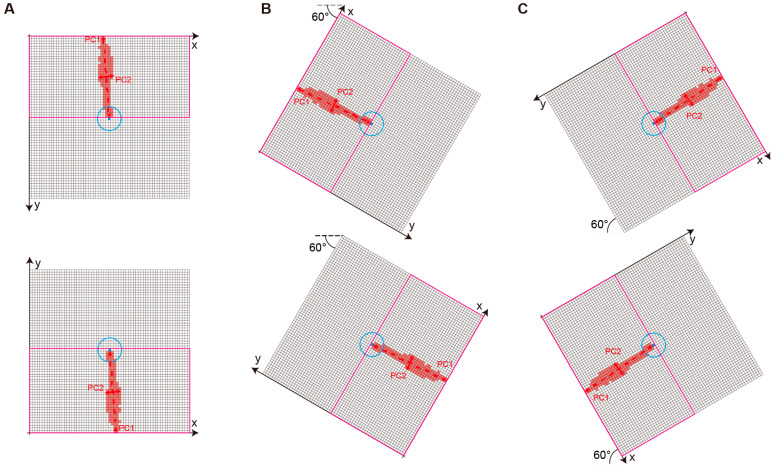
Schematic diagram of the tail presence determination algorithm for sperm candidates. The cyan dot represents the centroid of the sperm head candidate, and the cyan circle indicates the approximate range of the sperm head. Red clusters represent tail candidate pixels, and PC1/PC2 denote the principal axes computed from PCA. (**A**) Example with vertical tail orientation; (**B**,**C**) examples where the tail is diagonally oriented (~60°), and the region is rotated to align the tail vertically. This alignment allows consistent application of the tail detection logic regardless of the original direction.

**Figure 7 bioengineering-12-00601-f007:**
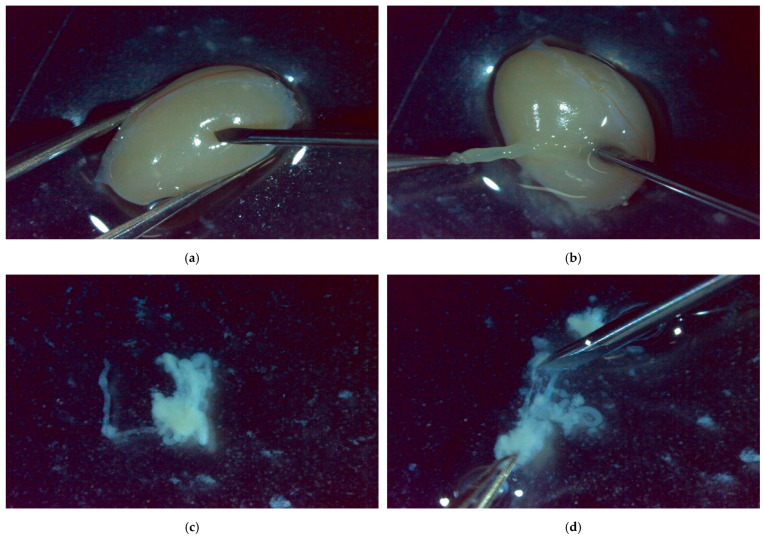
The procedure for collecting seminiferous tubule tissues from a mouse testis. (**a**) Incising a part of the testis with a micro scalpel. (**b**) Collecting a part of the seminiferous tubules from the incised area. (**c**) The collected seminiferous tubules. (**d**) Fragmenting the collected seminiferous tubules with a micro scalpel and collecting the seminiferous tubule tissues.

**Figure 8 bioengineering-12-00601-f008:**
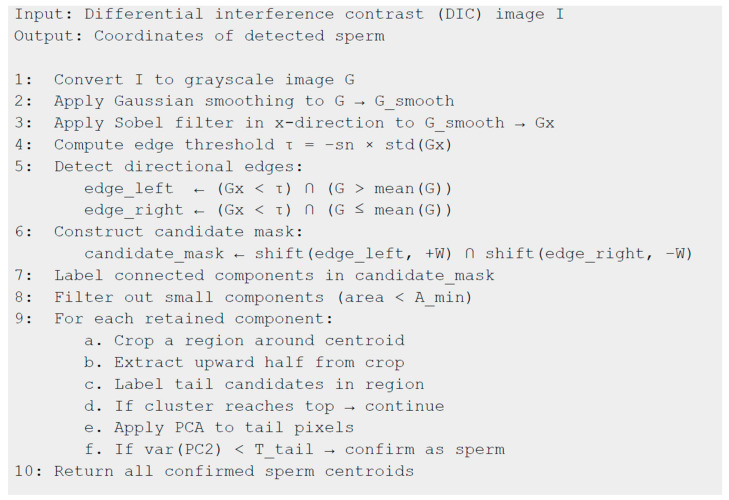
SD-CLIP main steps.

**Figure 9 bioengineering-12-00601-f009:**
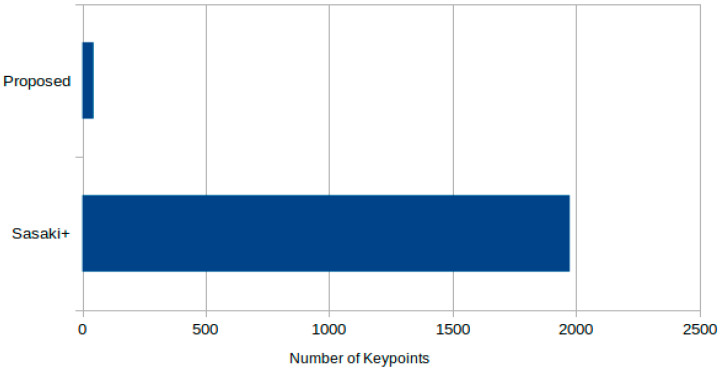
Comparison of the number of sperm candidate detection points between the traditional method and the proposed method.

**Figure 10 bioengineering-12-00601-f010:**
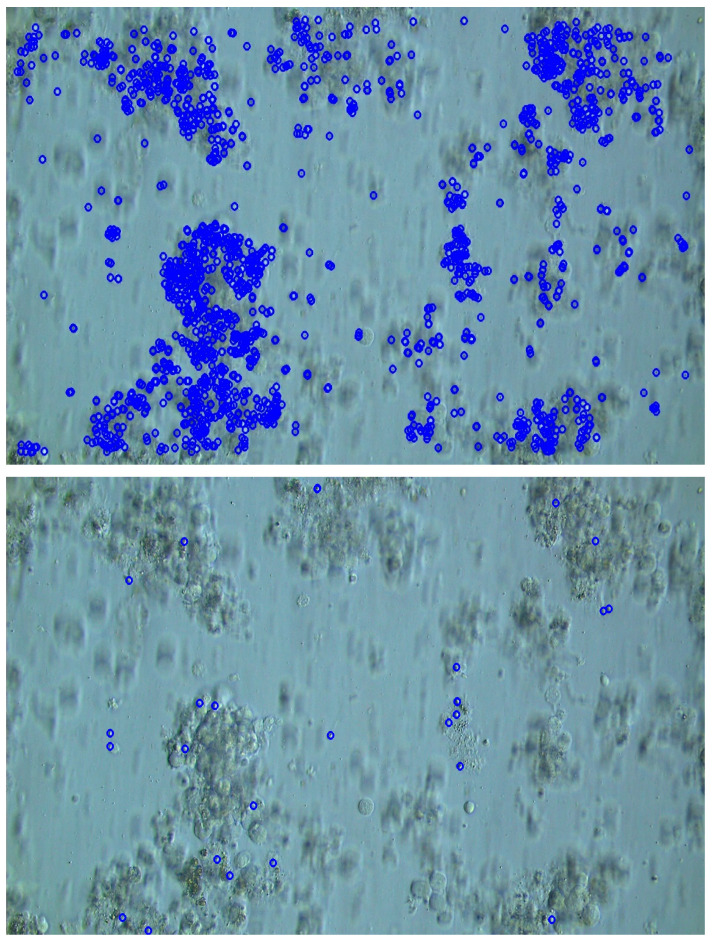
Comparison of sperm candidate detection results. The proposed method (**below**) shows fewer candidate points compared to Sasaki et al.’s method (**above**). Furthermore, in both methods, the sperm head located near the center is detected as a candidate.

**Figure 11 bioengineering-12-00601-f011:**
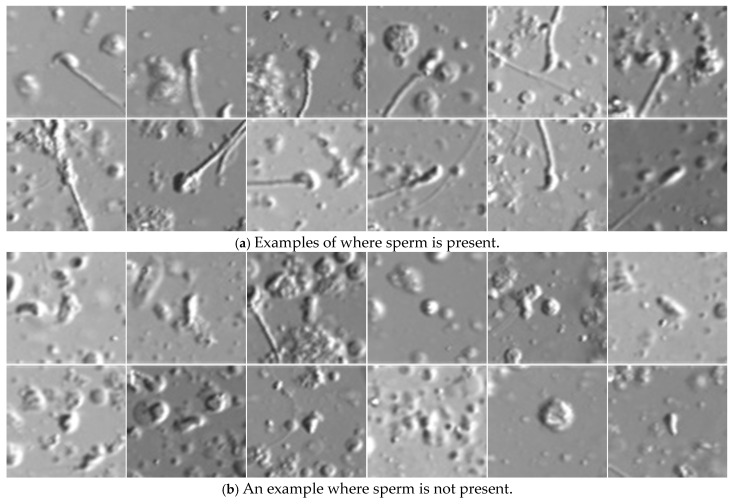
Examples of cells detected as sperm candidates. The top row (**a**) shows examples labeled as sperm present, while the bottom row (**b**) shows examples labeled as sperm absent. In this study, an image is labeled as “sperm present” only if it contains both a fully visible sperm head and at least part of the tail. If either condition is not met, the image is labeled as “sperm absent”.

**Table 1 bioengineering-12-00601-t001:** Results of sperm determination using both Sasaki et al.’s method and the proposed method.

Proposed Method		Correct Answer Label		Total	Sasaki et al.’s Method		Correct Answer Label		Total
		With Sperm	No Sperm				With Sperm	No Sperm	
Judgment result	With sperm	83	27	110	Judgment result	With sperm	7544	11,879	19,423
	No sperm	59	486	545		No sperm	1127	27,717	28,844
Total		142	513		Total		8671	39,596	

**Table 2 bioengineering-12-00601-t002:** A table comparing the accuracy of the proposed method with that of [7].

	True Positive Rate	False Positive Rate	True Negative Rate	False Negative Rate
Proposed Method (SD-CLIP)	58%	5%	95%	42%
Sasaki et al. (MB-LBP)	87%	30%	70%	13%

**Table 3 bioengineering-12-00601-t003:** Confusion matrix for sperm detection results.

		True Label	
		Sperm	No Sperm
Prediction	Sperm	83	27
	No Sperm	59	486

## Data Availability

The original contributions presented in this study are included in the article. Further inquiries can be directed to the corresponding author.

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
