# Peer review of "Fast and Accurate Sperm Detection Algorithm for Micro-TESE in NOA Patients"

_bioengineering, 2025, doi:10.3390/bioengineering12060601_

Round 1

Reviewer 1 Report

Comments and Suggestions for Authors

This paper proposes a classical image processing based method for sperm detection from unstained DIC images, which is fast and lightweight. Overall, the paper provides a clear contribution by offering a image processing solution that is both more efficient and effective compared to other methods such as MB-LBP + AKAZE. Specific comments are as follows:

  1. It is suggested to describe main steps of the proposed SD-CLIP method with pseudocode, so that readers can more easily follow and reproduce the method.
  2. Data, software package or source code could be made available to help researchers more easily use the algorithms and make direct comparisons.
  3. The algorithm SD-CLIP contains steps such as edge detection, connected component analysis, and labeling clusters. The paper could benefit from quantitative ablation experiments, in order to justify the exact choice of algorithm used for each step.

Author Response

"Please see the attachment

Reviewer 2 Report

Comments and Suggestions for Authors

Reviewer 3 Report

Comments and Suggestions for Authors

In this article, the authors proposed an image processing method for DIC microscopy to detect sperms fast with low false positive rate. The method is clearly described. But there are some issues to be addressed.

  1. The authors compared the proposed method with the previous MB-LBP in Table 2. Apparently, the number sin this table are not correct. The sum of the true positive rate and the false negative is not 100%.

    The performance of each method can be adjusted by its parameters. Therefore, there is lack of information to compare these two methods as the author did in Table 2. In order to argue the proposed method is better than the previous MB-LBP, the authors need to vary the parameters to generate the ROC curve (true positive rate with respect to false positive rate) of each method and then compare the quality of the two ROC curves.

  2. The authors based their methods on the detection of right and left edges of the DIC images. In real imaging, sometime the optical systems are not perfectly aligned, and the alignment of bright edge and the dark edge may be tilted. In this case the darkest and brightest edges are not exactly left or right. One simple solution is to add an image alignment step as a pre-processing to rotate the raw image so that the dark and bright edges are exact left or right.

  3. The authors demonstrated the proposed method has lower false positive rate compared to MB-LBP. But it also has lower true positive rate. As mentioned in point 1, this might be an inadequate comparison as the ROC curves are not compared. Another possibility is that the proposed method could be too stringent to select candidate. It is possible that a better method may follow the MB-LBP in the candidate selection, but then follow the proposed PCA step.

Round 2

Reviewer 1 Report

Comments and Suggestions for Authors

The authors have addressed the concerns from the reviewers.

Reviewer 2 Report

Comments and Suggestions for Authors

All corrections/comments have been addressed